# Treatment of Pelvic and Spinal Bone Metastases: Radiotherapy and Hyperthermia Alone vs. in Combination

**DOI:** 10.3390/cancers16081604

**Published:** 2024-04-22

**Authors:** Jong-Hun Kim, Jin-Yong Shin, Sun-Young Lee

**Affiliations:** 1Division of Thoracic and Cardiovascular Surgery, Jeonbuk National University Hospital-Jeonbuk National University Medical School, Jeonju 54907, Republic of Korea; kim77jh@gmail.com; 2Research Institute of Clinical Medicine of Jeonbuk National University-Biomedical Research Institute of Jeonbuk National University Hospital, Jeonju 54907, Republic of Korea; psjyshin@gmail.com; 3Department of Plastic and Reconstructive Surgery, Jeonbuk National University Hospital-Jeonbuk National University Medical School, Jeonju 54907, Republic of Korea; 4Department of Radiation Oncology, Jeonbuk National University Hospital-Jeonbuk National University Medical School, Jeonju 54907, Republic of Korea

**Keywords:** radiotherapy, modulated electrohyperthermia (mEHT), brief pain inventory (BPI), breakthrough pain (BTP), pain relief

## Abstract

**Simple Summary:**

Painful bone metastases significantly impact patients’ quality of life (QoL). Traditional treatments like morphine-equivalent medication (MeM) and local radiotherapy (RT)), but these interventions are not always successful. More recently, hyperthermia (HT) has been applied to complement RT and MeM, and this complex approach has shown promising synergistic results. Modulated electrohyperthermia (mEHT) combined with RT has shown promising results. This retrospective study with pelvic and spinal bone metastases compares the effects of RT and mEHT alone and in combination (RT + mEHT). RT + mEHT yielded significantly better results in terms of the above parameters. Clinically, mEHT has a lower risk of adverse thermal effects, and due to its efficacy, mEHT can be used to treat RT-resistant lesions.

**Abstract:**

Painful pelvic and spinal bone metastases are a considerable challenge for doctors and patients. Conventional therapies include morphine-equivalent medication (MeM) and local radiotherapy (RT), but these interventions are not always successful. More recently, hyperthermia (HT) has been applied to complement RT and MeM, and this complex approach has shown promising synergistic results. The objective of our study was to present the results of RT combined with a special kind of HT (modulated electrohyperthermia, mEHT), in which some of the thermal effect is contributed by equivalent nonthermal components, drastically reducing the necessary power and energy. This retrospective study included 61 patients divided into three groups with pelvic and spinal bone metastases to compare the effects of RT and mEHT alone and in combination (RT + mEHT). A detailed evaluation of pain intensity, measured by the brief pain inventory score, MeM use, and breakthrough pain episodes, revealed no significant differences between RT and mEHT alone; thus, these individual methods were considered equivalent. However, RT + mEHT yielded significantly better results in terms of the above parameters. Clinically, mEHT has a lower risk of adverse thermal effects, and due to its efficacy, mEHT can be used to treat RT-resistant lesions.

## 1. Introduction

Pain is the most frightening sign of cancer. It deeply impacts patients’ lives, often more than cancer [1,2]. More than 50% of advanced malignancies transform into bone metastases [3], and there are a variety of primary tumors [4]. Bone metastasis is a consequence of the complex exchanges of tumor cells with osteoblasts, osteoclasts, stromal cells, and inflammatory cells near bone [5], resulting in the transmission of pain signals to the cerebral cortex for perception. Bone metastasis is a severe, devastating issue with several adverse effects [6] involving bone remodeling, osteolytic lesion formation, hypercalcemia, cachexia, and bone fractures due to bone matrix degradation [7]. An estimated 10% of all back pain due to metastases is caused by spinal instability [8]. Spinal cord compression by vertebral collapse and pelvic collapse may prevent patients from moving, resulting in permanent bed rest.

Disseminated cancer cells cannot destroy bone; instead, they support several inflammatory mediators involved in a positive feedback loop of bone destruction [9]. The mechanism of bone metastasis is complex [10].

Furthermore, highly acidic necrotic tissues can form in the tumor mass due to insufficient nutrients and oxygen during intensive growth, decreasing the extracellular pH in the bone tumor microenvironment (TME) [11]. Additionally, growing tumors can cause painful mechanical compression of the spinal cord [12]. The mechanical influence depends on the position of the tumor lesion, i.e., extradural, intradural–extramedullary, or intramedullary.

Pain is a frequent cause of worsening quality of life (QoL) and is the most feared and troublesome symptom of this disease. Due to hardening of the joints, pain may increase at night [13]. A large number (40–80%) of patients with bone metastases experience transient episodes of breakthrough pain (BTP), which are severe, unbearable pain spikes that exceed the controlled ongoing background pain [14]. BTP can be triggered 4–6 times per day, and this excruciating, debilitating pain can cause dramatically reduced QoL [15].

Painful bone metastases are conventionally treated by radiotherapy (RT) [16], in which highly focused radiation beams are directed to the sites of bone pain, frequently in the vertebral column. The World Health Organization (WHO) guidelines for treating cancer-related pain recommend single-dose, low-fractionated, external-beam RT for bone metastatic cancer-related pain [17].

RT can provide several benefits, including relief from bone metastatic pain [18], when applied according to the American Society of Radiation Oncology (ASTRO) guidelines for bone metastases [19,20]. Approximately 60–85% of patients report partial pain relief with RT, while 15–58% report complete pain relief [21,22]. Pain relief from external-beam RT can last between one and six months. While there have been studies on fractionating the RT dose [23,24], fractioning does not appear to modify the results, as the same excellent results regarding pain control can be achieved with a single dose of 8 Gy or 4 Gy/fraction in 5 or 6 sessions and 3 Gy/fraction in 10 sessions [19].

A comprehensive treatment plan, which may include a combination of RT and other therapies, such as hyperthermia (HT) [25], pain medications, and supportive care, is typically tailored to the individual’s specific needs to provide optimal pain relief and improve QoL. HT is essentially a complementary therapy, mostly used in conjunction with RT. The nonionizing radiation in HT decidedly sensitizes the body to the ionizing radiation in RT [26,27,28], as supported by a broad spectrum of evidence [29,30,31,32,33].

HT is frequently applied with RT as a complementary treatment modality to relieve pain [34], especially pain from bone cancer [35]. The rationale is based on complex synergy between the thermal effect and the effect of RT [36]. Despite the differences between ionizing (RT) and nonionizing (HT) processes, the two methods result in synergistic therapeutic effects when applied in combination, and not only in cases of bone pain [37]. HT supports RT by thermosensitizing [38] and reoxygenating the target [39], and active arrest of the cell cycle in different phases by RT and HT results in a synergistic effect. RT is most effective against cells in mitosis, while moderate HT predominantly acts on cells in the S phase [40] and blocks the G1/S and G2/M checkpoints [41]. Various molecular processes facilitate the efficacy of RT [42], such as the thermally induced decrease in DNA-dependent protein kinase activity [43]. Additionally, HT increases the oxygen supply by increasing blood perfusion, which supports the repair of RT-induced DNA breaks [44,45]. Furthermore, the homogeneously high temperatures of HT could block enzyme activity [46], arrest the activity of DNA-repairing enzymes, and optimize the cellular degradation of malignant cells [47]. HT may increase perfusion in tumors and thus increase oxygenation and improve drug delivery [48].

However, both methods have disadvantages and unwanted adverse effects, including acute nausea/vomiting and diarrhea. Side effects can also affect normal tissues through acute (early, consequential) and late adverse processes [49]. RT may damage the spinal cord, causing loss of strength, paralysis, or problems with bowel or bladder control, and nerve damage can cause the sensation of an electric shock. RT may also damage the spine, causing a reduction in the patient’s height or the formation of a curved spine, increasing the risk of fracture. Radiation in the thoracic area may cause esophagitis. RT of the spinal cord may cause acute transient myelopathy [49] or later affect lower motor neuron syndrome, telangiectasias, and subsequent hemorrhage [50]. Late effects can also lead to progressive myelopathy, resulting in irreversible neurological deficits.

HT also has adverse effects, including erythrogenic skin reactions, erythema, skin burns, or deeper adipose burns in subcutaneous layers. Patients may complain of pain during local HT. The intensive pain and weak condition of patients with bone metastases may limit the relatively long (30+ min) duration of treatment with intensive thermal effects, reducing the quality HT [51,52].

Both RT and HT involve the development of heat shock proteins (HSPs) (HT [53,54] and RT [55]), which may lead to resistance to the combined treatments. Suboptimal homogeneous mass heating may enhance the activity of DNA repair enzymes and limit DNA destruction by RT. Furthermore, hypoxia potentially resulting from HT could also limit the efficacy of RT, while optimal HT application may act in the opposite manner, facilitating RT with intensive blood perfusion [39] and thermosensitization [38].

The modulated electrohyperthermia (mEHT) method [56] applies definite heating in the fever range [57] completed with nonthermal bioelectromagnetic effects [58]. This method selectively excites transmembrane proteins in malignant cells [59]. The selectivity of mEHT is dependent on physiologic heterogeneities, such as how malignant cells differ thermally and electrically from healthy cells. Malignant cells have a higher metabolic rate and greater autonomy, and the radiofrequency current is driven by the high ionic density and dielectric permittivity in the TME [60]. Furthermore, healthy cells have homeostatic electrolyte concentrations in different regions, while cancer cells modify their composition and pH, which promotes tumor selectivity. These selective properties also apply to bone metastases.

The mEHT method addresses the disadvantages of high power levels. Heating by mEHT does not target the whole tumor mass; instead, individual malignant cells are affected [61]. This approach avoids overly intensive feedback from homeostatic regulation and reduces the risk of adverse thermal effects [62]. Due to the ionizing radiation, RT also acts selectively, mostly by targeting the DNA of malignant cells and causing cell death. This nanoscale targeting is similar to that of the nonionizing radiation of mEHT [36], which targets the membrane rafts of malignant cells via both thermal and nonthermal processes [63] and triggers programmed cell death [64].

The first phase III clinical study of RT (10 × 3 Gy) + HT (4 × 40 min) (n = 29) compared to RT alone (n = 28) [65] was an important step forward in clarifying the clinical value of this treatment combination. The highest intratumorally measured temperature was 41.9 ± 1.2 °C. The median power in the capacitively coupled Thermotron device was 559.3 W/treatment, ranging from 300 to 1250 W/treatment; the power was limited by patient discomfort. Pain was measured using the standard brief pain inventory (BPI) [66,67], with a score of zero indicating a complete response (CR). The CR rate differed significantly between RT + HT (37.9%) and RT alone (7.1%). Furthermore, the duration of pain relief (DPR) was significantly longer in the RT + HT arm. Although the results were significant, the small number of patients in the two randomized groups limited the statistical power [68,69].

The objective of our study was to present the results of RT combined with a special kind of HT (mEHT) [36,63], in which nonthermal factors also play a role [70,71]. This method requires less power, providing patients with better comfort and lowering the risk of thermal or combined side effects.

## 2. Materials and Methods

### 2.1. Patient Selection

In the present retrospective study, we selected patients with confirmed pelvic and spinal bone metastases irrespective of their origin, with histological or clinical confirmation provided by computed tomography (CT), magnetic resonance imaging (MRI), or positron emission tomography (PET)/CT. The inclusion criteria were as follows:Age between 20 and 85 years.Life expectancy > 6 months.ECOG status score < 3.Lesion size < 25 cm due to the limitations of the irradiated and heated fields.Severe pain (BPI > 4) for more than 24 h.Solitary metastasis.

Sixty-one patients were eligible according to the inclusion criteria. This cohort was divided into three groups: (1) RT alone (n = 21); (2) mEHT alone (n = 20); and (3) combined RT + mEHT (n = 20). The inclusion period was from January 2018 to December 2022. Figure 1 illustrates the grouping of the patients, and Table 1 shows the patients’ baseline characteristics.

### 2.2. Radiation Therapy

The RT protocol consisted of 30 Gy, with 3 Gy/fraction, administered five times a week for two weeks. Diagnostic imaging (CT, MRI, or PET/CT) was performed with a slice thickness of 5 mm based on a computer simulation. The target volume was considered to include the metastatic bone lesion and the neighboring soft tissue. To avoid untreated micrometastases, the clinical target volume included an additional contour of at least 20 mm around the complete imaging volume, and the planning target included 5 mm of extra space around the complete clinical target volume. Radiation therapy was performed using 10 MV photon energy. The planned target volume was generated with a three-dimensional margin of 5 mm around the clinical target volume. All patients underwent three-dimensional conformal radiotherapy. Three-dimensional conformal RT was generated using a Clinic IX (Varian Medical System, Palo Alto, CA, USA) machine by the EClipsetreatment planning system (Varian Medical System, Palo Alto, CA, USA).

### 2.3. Hyperthermia

The mEHT treatment was performed with an EHY2000+ device (Oncotherm, Troisdorf, Germany) 3 times/week for 4 weeks. A 2D projection of the 3D simulation was used to facilitate accurate positioning of the 30 cm circular mEHT applicator. Prior to modulated-electrohyperthermia, all patients underwent a two-dimensional simulation by Acuity (Varian Medical System, Palo Alto, CA, USA). The treatment field encompassed the mass with a 3 cm margin in the X and Y directions. The body temperature, blood pressure, and pulse rate of each patient were measured prior to, during, and following treatment. Body temperature was measured using an infrared ear thermometer (Infrared Thermometer IRT 4020; Braun GmbH, Kronberg, Germany), and the temperature of the treatment area skin surface below the circular upper electrode probe was measured using a non-contact infrared thermometer transmitter (Thermo Checker DT-060; Easytem Co., Ltd., Siheung, Republic of Korea). Tumor coverage by the applicator and patient comfort were optimized during positioning. Most patients (60%) were treated in the supine position. According to the mEHT guidelines [72], the session time was 60 min. A step-up protocol was applied to start at 80 W for 10 min, increase to 100 W for 10 min, and subsequently increase to 120 W for 10 min. The power in the final 30 min was limited by the tolerance of the patient and regulated by the discomfort of the patient but did not exceed 150 W. The average optimal energy was 433 kJ (Table 2). The energy applied was considered the treatment dose [73]. The temperature achieved at the membrane of malignant cells by the applied protocol was higher than 41 °C [36], while the average expected temperature was ≈39.5 °C, as measured in the pelvis (cervical tumors) [74].

### 2.4. Combined Radiotherapy and Hyperthermia

The protocol for the combined treatment aimed to achieve the greatest synergistic effect. The two treatments began on the same day; daily RT was combined with mEHT every second day within 1 h after RT for the first two weeks, and then, mEHT was applied alone for two weeks.

The results of treatment were evaluated by the pain intensity according to the BPI score. The response rate (RR) was measured immediately and four weeks after treatment, while the DPR was recorded in weeks. Responses were measured at the treated sites. A CR was defined as a BPI score of 0 without an increase in analgesic intake measured in daily oral morphine equivalents. According to the international consensus [75,76], a partial response (PR) was defined as a decrease in the BPI score of more than two without an increase in the use of analgesics. Progressive pain (PP) was defined by an increase in the BPI score of two or more points above baseline at the treated site, with stable analgesic use or an increase of 25% or more in daily oral morphine equivalent intake with a stable BPI score or a BPI score one point above baseline. A PR or PP outcome was considered to indicate no change (NC), while the CR + PR rate was considered to indicate the overall RR, representing remarkable relief from bone pain. Evaluations were performed immediately and 4 weeks after treatment (Table 3).

### 2.5. Statistical Analysis

To compare the differences between the groups, a student *t* test was used. The Kaplan–Meier method, a log-rank test, and the Cox’s proportional hazard model were used for comparing the between-group differences of the CR and PR rates. The statistical software SPSS version 22.0 (IBM, Armonk, NY, USA) was used for performing statistical analyses. Statistically significant was defined as a *p* value of < 0.05.

## 3. Results

The results of the present investigation support these earlier observations of reduced pain from bone metastases (Table 4). The significance of the results is shown in Figure 2.

Table 5 shows the RRs collected immediately and four weeks after treatment according to the BPI score.

The RR was nearly complete in all three treatment groups (Table 6).

The development of pain measured by the BPI score during the follow-up period differed among the groups (Table 7). In this follow-up evaluation, the BPI score obtained immediately after treatment was considered the baseline.

The notable decrease in the BPI score after the combined treatment compared to either treatment alone shows the advantage of the combined therapy (Figure 3).

Similar to the BPI score, the DPR also decreased, favoring the combined treatment (Figure 4).

A comparison of the two types of bone metastases revealed similar results for spinal and pelvic bone metastases (Figure 5).

The DPR was also similar between the two types of bone metastases (Figure 6). No significant differences were observed between spinal and pelvic lesions.

At four weeks after treatment, significant reductions in the BPI score and the BTP frequency were observed in the combined treatment group (Figure 7).

Nonparametric Kaplan–Meier estimation showed significantly better pain relief over time with the combined treatment than with RT or mEHT alone, with no significant difference between the two monotherapies (Figure 8).

The overall RR calculated according to international consensus guidelines [20] demonstrates the excellence of all three methods (Figure 9).

The adverse effects were predominantly caused by RT. Patients treated in the thoracic spine experienced esophagitis (Radiation Therapy Oncology Group [RTOG] grade I–II), but symptoms disappeared within 4 weeks. No other side effects were observed; mEHT did not produce any side effects other than a mild hot sensation in the treatment area, the discomfort of which limited the applied power.

In the modulated electrohyperthermia alone group, body temperature before the treatment ranged from 36.4 to 36.9 °C (mean 36.5 °C) and that after treatment ranged from 36.9 to 37.8 °C (mean: 37.4 °C), indicating an average rise of 0.9 °C. The temeprature of the treatment area skin surface underneath the upper electrode before heating ranged from 35.2 to 36.1 °C (mean 35.3 °C) and it increased to 39–40.5 °C (mean: 40.1 °C). In the radiotherapy combined modulated electrohyperthermia group, body temperature before the treatment ranged from 36.2 to 36.7 °C (mean: 36.4 °C) and that after treatment ranged from 36.8 to 37.8 °C (mean: 37.4 °C), indicating an average rise of 1 °C. The temeprature of the treatment area skin surface underneath the upper electrode before heating ranged from 35.0 to 36.1 °C (mean: 35.2 °C) and it increased to 39–40.4 °C (mean: 40 °C), indicating an average rise of 0.8 °C.

## 4. Discussion

The applied treatment is a novel approach consisting of a conventionally accepted RT protocol [20] combined with a special type of HT, which includes nonthermal and thermal components [36,63]. The increased efficacy of this approach despite the lower temperature (in the fever range) was expected according to theory and preclinical investigations [70,77].

Various temperature-dependent mechanisms may be described by the Arrhenius law [78]. In the Arrhenius-based Henrique–Moritz model [79], cell damage is necrotic in nature and can be modeled by parallel, dominantly irreversible chemical reactions. However, robust cellular repair mechanisms may block cell death in some cells with heat resistance [80]. Necrotic processes can become painful before burns occur, as the dose threshold for discomfort is 10–100 times lower than the thermal dose needed for necrosis [81].

Thermal energy does not activate all cell-damage processes or simply result in immediate necrosis, and the actual effects can vary. In a simple case when electromagnetic heating is applied, we may suppose at least one deviation in the effects out of the following:The dominant effect may result from temperature changes, which depend only on the absorbed energy (specific absorption rate, SAR) irrespective of the energy source.Some of the effects could also depend on the mode of energy delivery, that is, the actual electromagnetic field [82].

Interactive forces cause a lasting impact on ionic charges and dipoles of the constructive elements of biomatter. While these effects are thermally induced, they are not necessarily thermal and behave as at least first-order phase transitions, with the so-called “latent heat”. This means that, after reaching the transition temperature, the value does not change until the transition process is completed. In this interval, the temperature will be independent from the SAR.

Electromagnetic interactions create nonthermal effects [83], and the balance of thermal and nonthermal effects optimizes the synergy between RT and mEHT [36]. The applied power and absorbed energy are shared by thermal and nonthermal processes, offering the clinical advantage of a moderate thermal component. This optimum balance reduces the risk of adverse thermal effects, inducing patient discomfort, while exerting numerous antitumoral immunogenic effects, killing tumor cells via programmed cell death and leading to tumor-specific immune reactions [84].

Previous case reports have demonstrated the success of mEHT in treating various bone metastases:A cholangiocarcinoma metastasis in pelvic bone was treated with mEHT (12×) + RT (15 × 3 Gy) combined with the checkpoint inhibitors Yervoy and Opdivo [85].A solitary non-small cell lung carcinoma (NSCLC) bone metastasis in the left hip was treated with mEHT (60 min, 70 W) and bevacizumab chemotherapy [86].A long-term response was achieved in patients with extensive bone metastases from prostate cancer treated with mEHT (100 × 50 min, 95 W average) + RT (30 × 1.5 Gy) combined with antitumor viral therapy [87].Bone metastasis from NSCLC was successfully treated with mEHT (22×) + RT (10 × 3 Gy) combined therapy [88].An observational study of 19 patients treated with standard palliative pain therapy (PaT, n = 10) or combined PaT + mEHT (n = 9) [89] included a broad spectrum of bone metastases in the following locations: the ribs (n = 3); the pelvis (n = 3); the sternum (n = 3); solitary and lumbar spine + pelvis (n = 10); ribs + pelvis (n = 2); sternum + collarbone + ribs (n = 1); and multiple bone metastases originating from mesothelioma (n = 1) and the breasts (n = 6), the lung (n = 4), the prostate (n = 4), the colon (n = 1), the oropharynx (n = 1), the ovaries (n = 1), and the pancreas (n = 1). The mEHT treatment time (60 min) and power (80 → 150 W) were gradually increased over three sessions/week, with standard PaT provided over a few weeks. The combined treatment results were compared to those of PaT alone, showing the strong superiority of the combined treatment in terms of the visual analog scale (VAS) pain score and Eastern Cooperative Oncology Group (ECOG) performance status score evaluated after half a year.

Conventionally, RT is considered an effective treatment modality for bone metastases, reducing pain in more than 50% of patients [76] and providing 12 weeks of pain relief in less than 30% of patients [90]. HT combined with RT has long been acknowledged as a treatment modality for malignant tumors, and the first phase III clinical trial [65] demonstrated the excellent capacity of complementary HT with RT to reduce pain from bone metastases. The results of the present comparative data are not different from those of the previous report. Relief of pain intensity was better in complementary HT with RT than in monotherapy, and pain relief lasted longer in the HT + RT group. Moreover, HT, as a complementary adjuvant, has immunogenic activity [84,91].

The synergy of RT and nonthermal processes is less pronounced immediately after treatment, as biological effects that require time to fully manifest are triggered. A comparison of the pain immediately (Figure 5B) and four weeks (Figure 5C) after treatment demonstrates well this time lag, with significantly better results for both spinal and pelvic lesions long after treatment. Notably, no such difference was observed for RT or mEHT alone. At four weeks after treatment, significant reductions in the BPI score and the BTP frequency were observed in the combined treatment group, as well as a slight but not significant reduction in the frequency of morphine equivalent pain medication intake (Figure 7).

The detailed evaluation of the BPI score, the pain medication intake, and the BTP frequency showed no significant differences between RT and mEHT alone but significantly better results for combined RT + mEHT. Clinically, it is remarkable that mEHT carries a lower risk of adverse thermal effects (e.g., no erythema, burns, discomfort, nausea/vomiting, etc.), and patients with complaints during the treatment may immediately limit the already much lower power than that in conventional HT without reducing the therapeutic benefits. The main limitation of the present study is its retrospective nature; thus, further research following a randomized study design is warranted.

## 5. Conclusions

The results of this three-group investigation comparing the effects of RT and mEHT alone and in combination show the feasibility of mEHT for painful bone metastases in the pelvic and spinal areas.

## Figures and Tables

**Figure 1 cancers-16-01604-f001:**
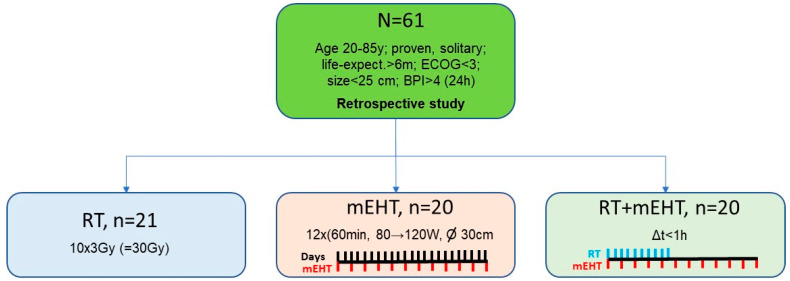
Grouping of patients according to treatment protocol.

**Figure 2 cancers-16-01604-f002:**
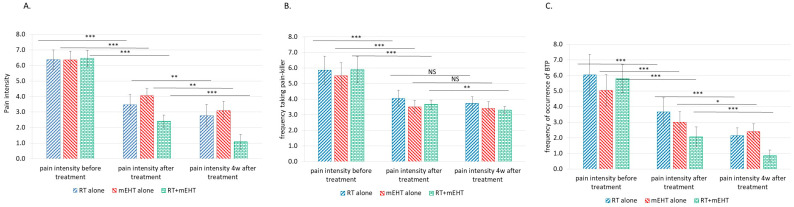
Changes in the average pain intensity score (**A**), analgesic frequency (**B**), and BTP frequency (**C**) in the different patient groups at different times; * (*p* < 0.05), ** (*p* < 0.005), and *** (*p* < 0.00005); NS, no significance.

**Figure 3 cancers-16-01604-f003:**
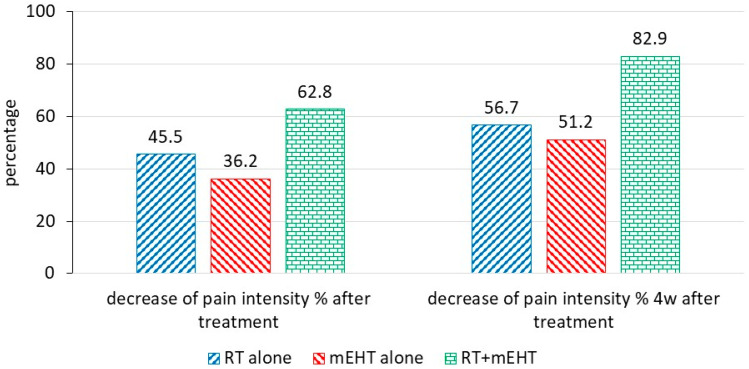
Percent decrease in pain intensity at two time points after treatment in the different patient groups.

**Figure 4 cancers-16-01604-f004:**
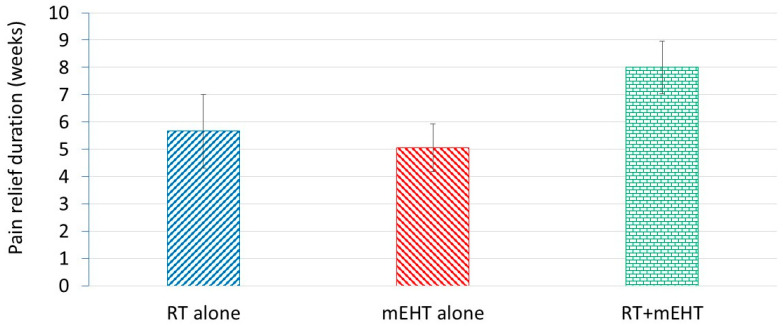
DPR (in weeks) for patients in the different groups.

**Figure 5 cancers-16-01604-f005:**
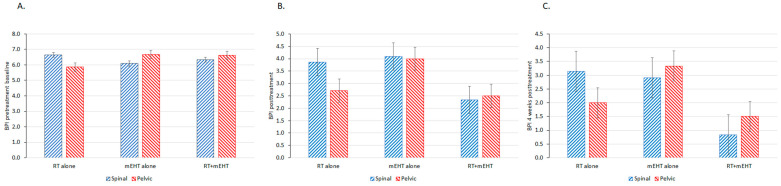
BPI scores for patients with bone metastases in the pelvis and spine: (**A**) at baseline; (**B**) immediately after treatment; and (**C**) and four weeks after treatment.

**Figure 6 cancers-16-01604-f006:**
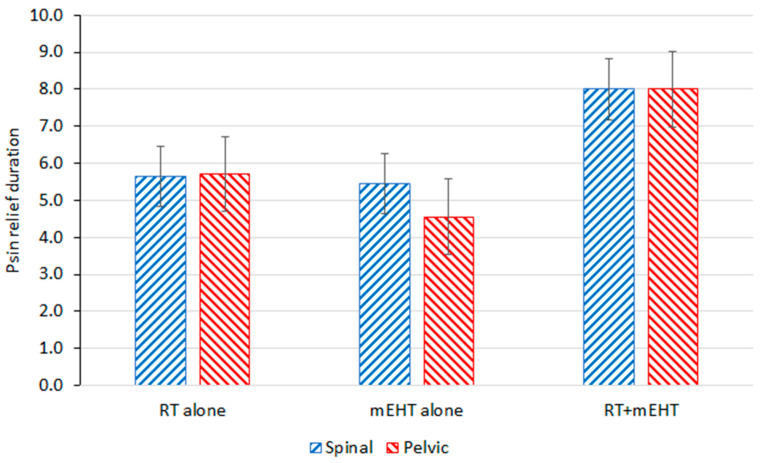
DPR after treatment for pelvic and spinal bone metastases.

**Figure 7 cancers-16-01604-f007:**
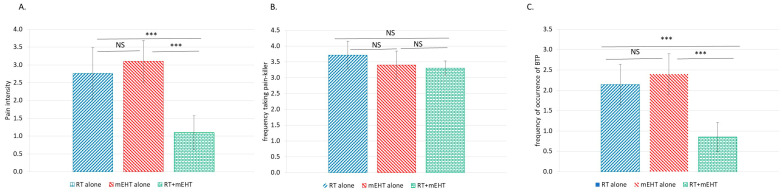
Changes in pain intensity (**A**), analgesic frequency (**B**), and BTP frequency (**C**) in the different patient groups four weeks after treatment; *** (*p* < 0.00005); NS, no significance.

**Figure 8 cancers-16-01604-f008:**
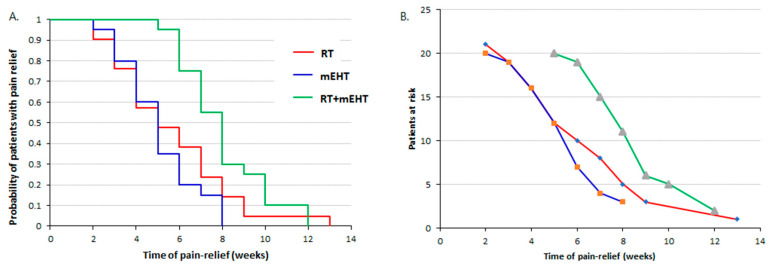
Nonparametric Kaplan–Meier estimation of the DPR in the different patient groups: (**A**) probability of patient pain relief over time and (**B**) patients at risk over time.

**Figure 9 cancers-16-01604-f009:**
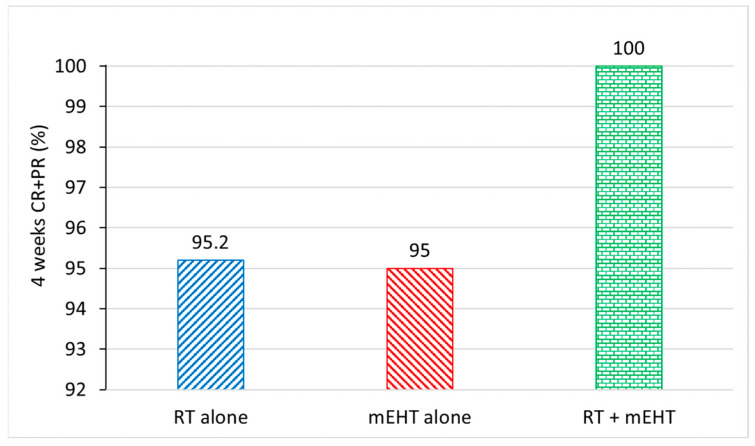
Remarkable pain relief in different groups.

**Table 1 cancers-16-01604-t001:** Patient characteristics at baseline.

Characteristics		RT Alone	mEHT Alone	RT + mEHT
**Patients’ number**		21	20	20
**Age(y) (mean ± SD)**		60.8 ± 8.9	62.0 ± 8.9	63.3 ± 8.8
**Sex (n, %)**				
	Male	15 (71.4)	11 (55.0)	13 (65.0)
	Female	6 (28.6)	9 (45.0)	7 (35.0)
**Primary cancer site**				
	Breast	4 (19.0)	5 (25.0)	3 (15.0)
	Cervix	0 (0.0)	2 (10.0)	1 (5.0)
	Lung	10 (47.6)	5 (25.0)	7 (35.0)
	Prostate	5 (23.8)	3 (15.0)	5 (25.0)
	rectum	2 (9.5)	5 (25.0)	4 (20.0)
**Metastasis location**				
	Pelvic	7 (33.0)	9 (45.0)	8 (40.0)
	Spinal	14 (67.0)	11 (55.0)	12 (60.0)
**BPI**		6.4 ± 1.3	6.4 ± 1.1	6.5 ± 1.1
**Frequency of pain killer**		5.9 ± 1.9	5.5 ± 1.7	5.9 ± 1.7
**BTP**		6.0 ± 2.7	5.1 ± 2.0	5.8 ± 1.9

BPI: brief pain inventory; BTP: breakthrough pain.

**Table 2 cancers-16-01604-t002:** Average mEHT treatment power/energy values.

Characteristics	mEHT Alone	RT + mEHT
Starting power (W)	80 ± 0	80 ± 0
Mean of final power (W)	143 ± 7.1	141.5 ± 7.9
Mean power (W)	110.8 ± 23.6	110.4 ± 23.2
Mean energy (kJ)	433.2 ± 12.7	430.5 ± 13.9

mEHT: modulated electrohyperthermia; RT: radiotherapy.

**Table 3 cancers-16-01604-t003:** Responses categorized by pain [76].

Definition/Response	CR	PR	NC	PD
**The drop in BPI pain score**	Complete vanishes	Drop by 2 or more	Constant or changes of ±1	Grows by 2
**With analgesic use (oral daily morphine equivalent)**	No increase	No increase	Increase of less than 25%	Stable or increases

CR: complete response; PR: partial response; NC: no change; PD: progressive disease.

**Table 4 cancers-16-01604-t004:** Average pain intensity score, analgesic frequency, BTP frequency, and DPR after treatment in the three groups. (Numbers in parentheses are standard deviations).

Characteristics	RT Alone	mEHT Alone	RT + mEHT
**Pain intensity (average)**			
Before treatment	6.4 ± 1.26	6.35 ± 1.2	6.45 ± 1.0
After treatment	3.5± 1.2	4.05 ± 1.0	2.4 ± 1.6
Four weeks after treatment	2.8 ± 1.4	3.1 ± 0.8	1.1 ± 1.0
**Frequency taking analgesics (average)**			
Before treatment	5.9 ± 1.8	5.5 ± 1.6	5.9 ± 1.6
After treatment	4.0 ± 1.0	3.5 ± 0.8	3.65 ± 0.6
Four weeks after treatment	3.7 ± 0.8	3.4 ± 0.8	3.3 ± 0.4
**Frequency of BTP occurrence (average)**			
Before treatment	6.0 ± 2.6	5.05 ± 2.0	5.8 ± 1.8
After treatment	3.7 ± 1.8	3.0 ± 1.4	2.05 ± 1.2
Four weeks after treatment	2.1 ± 1.2	2.4 ± 1.0	0.85 ± 0.8
**Pain-relief duration four weeks after treatment (average)**			
Four weeks after treatment	5.7 ± 2.6	5.05 ± 1.8	8.0 ± 2.0

RT: radiotherapy; mEHT: modulated electrohyperthermia, BTP: breakthrough pain.

**Table 5 cancers-16-01604-t005:** RRs collected immediately and four weeks after treatment in the three distinct groups.

Response Compared to Pretreatment Baseline	CR	PR	NC	PD
**Immediate**				
RT alone	2	18	1	0
mEHT alone	0	18	2	0
RT + mEHT	0	20	0	0
**Four-week follow-up**				
RT alone	3	17	0	1
mEHT alone	1	18	1	0
RT + mEHT	7	13	0	0

CR: complete response; PR: partial response; NC: no change; PD: progressive disease; RT: radiotherapy; mEHT: modulated electrohyperthermia.

**Table 6 cancers-16-01604-t006:** Number of patients with a marked decrease (≥2 points) in the BPI score (CR + PR) and an improvement in QoL. (Numbers in parentheses are percentages).

RR	CR + PR (%)
**Immediate**	
RT alone	20 (95.2)
mEHT alone	18 (90.0)
RT + mEHT	20 (100.0)
**Four-week follow-up**	
RT alone	20 (95.2)
mEHT alone	19 (95.0)
RT + mEHT	20 (100.0)

RR: response rate; CR: complete response; PR: partial response; RT: radiotherapy; mEHT: modulated electrohyperthermia.

**Table 7 cancers-16-01604-t007:** RRs in the follow-up period. The pain score immediately after treatment was considered the baseline. (Numbers in parentheses in the RR column are percentages).

Follow-Up Response Compared to that Immediately Post-Treatment	CR	PR	NC	PD	RR (%)
RT alone	3	3	14	1	6 (28.6)
mEHT alone	1	5	14	0	6 (30.0)
RT + mEHT	7	2	11	0	9 (45.0)

CR: complete response; PR: partial response; NC: no change; PD: progressive disease; RR: response rate; RT: radiotherapy; mEHT: modulated electrohyperthermia.

## Data Availability

Data are available upon request.

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
