# Peer review of "Treatment of Pelvic and Spinal Bone Metastases: Radiotherapy and Hyperthermia Alone vs. in Combination"

_cancers, 2024, doi:10.3390/cancers16081604_

Round 1

Reviewer 1 Report

Comments and Suggestions for Authors

The authors provide a paper about "Treatment of Pelvic and Spinal Bone Metastases: Radiotherapy and Hyperthermia Alone vs. in Combination".

The topic is important and certainly deserves attention howver there are few points that I would like the authors to address:

1) How were patients grouped? Who decided the different theraputic approach?

2) Were patients managed in a multidisciplinary setting?

3) In analysing results the authors should take into account the baseline features of the pain (patients who received only RT had a worse baseline)

4) In the figure the authos shoul use an homogeneous way to show results (figure 3 show the entire percentage whereas figure 8 only from 92 to 101 (???)

6) please remove the value of 101% in figure 8 

7) please provide the mean CTV size of the targets treated

Author Response

1) How were patients grouped? Who decided the different theraputic approach?

The treatment group for patients was determined by directly asking the patient for his or her opinion and whether there was pain in the area where radiation treatment was previously performed. When radiation therapy was requested for bone metastases, the patient was asked whether to use it in combination with hyperthermia. If he or she said yes, the combination treatment was performed. If the patient refused, the patient was treated with radiation therapy alone. If the patient's general condition made it difficult to administer radiation therapy, if there was pain in an area where radiation therapy had been previously performed, or if the patient refused radiation therapy, hyperthermia treatment was performed.

2) Were patients managed in a multidisciplinary setting?

For patients with bone metastases, multidisciplinary treatment is not yet available at the researcher's hospital. Treatment was carried out in collaboration with individual medical staff.

3) In analysing results the authors should take into account the baseline features of the pain (patients who received only RT had a worse baseline)

Thank you for pointing out. The reason why pain intensity was higher in the group of patients who received only radiation therapy was because radiation therapy is more effective in controlling pain than heat therapy, so a high proportion of patients with high pain intensity chose radiation therapy alone when explaining the treatment process. Better research results can be derived if the study is conducted through prospective random assignment, but because there are factors that impede the patient's quality of life, such as pain that has already metastasized, random assignment was not possible and the study was conducted through retrospective analysis. In the analysis, initial pain intensity was found to be higher in the radiation-only group In future studies, we will conduct prospective random assignment studies to reduce bias as much as possible.

4) In the figure the authos shoul use an homogeneous way to show results (figure 3 show the entire percentage whereas figure 8 only from 92 to 101 (???)

Thank you for your comment. We adjusted the scale to make the differences between the data easier to understand in figure 9 (number corrected).

6) please remove the value of 101% in figure 8

We removed the value of 101 %.

7) please provide the mean CTV size of the targets treated

In the explanation of radiation therapy and hyperthermia, setting the patient's treatment range was explained in more detail.

Radiation therapy

After immobilization and CT simulation, the target volume was defined after registration of the diagnostic magnetic resonance imaging, CT, or positron emission tomography/CT scan to the simulation CT scan (5-mm slice thickness). The affected bony lesions, including soft tissue parts, were delineated as the gross tumor volume. For coverage of areas with potential microscopic disease, a safety radical margin at least 20 mm around the gross tumor volume was defined as the clinical target volume. The planning target volume was generated with a 3-dimensional margin of 5 mm around the clinical target volume. Radiation therapy was performed using 10MV photon energy. All patients underwent three-dimensional conformal radiotherapy. Three-dimensional conformal RT was generated from a Clinic IX (Varian Medical System, Palo Alto. CA, USA) machine by the EClipse treatment planning system (Varian Medical System, Palo Alto. CA, USA).

Hyperthermia

The mEHT treatment was performed with an EHY2000+ device (Oncotherm, Troisdorf, Germany) 3 times/week for 4 weeks. A 2D projection of the 3D simulation was used to facilitate accurate positioning of the 30-cm circular mEHT applicator. Prior to modulated-electro hyperthermia, all patients underwent a two-dimensional simulation by Acuity (Varian Medical System, Palo Alto. CA, USA). The treatment field encompassed the mass with a 3 cm margin in the X and Y directions. The body temperature, blood pressure and pulse rate of each patient were measured prior to, during and after treatment. Body temperature was measured using an infrared ear thermometer (Infrared Thermometer IRT 4020; Braun GmbH, Kronberg, Germany) and the temperature of the treatment area skin surface below the circular upper electrode probe was measured using a non-contact infrared thermometer transmitter (Thermo Checker DT-060; Easytem Co., Ltd., Siheung, Korea). 

Reviewer 2 Report

Comments and Suggestions for Authors

General comments:

The paper tackles the effect on pain reduction caused by bone metastases after radiotherapy, hyperthermia or combined therapies. The paper requires rewriting and restructuring due to its repetitive nature (particularly the information in the Introduction), lack of important data regarding treatment techniques and quite chaotic presentation of results, which appear in all sections of the paper (instead of being presented only in the Results section). Also, the Methods includes information that should be either in the Introduction or in the Results. Below are my comments:

Introduction:

1.     The authors put too much emphasis on the pain induced by bone cancer, which becomes highly repetitive throughout the paragraphs. The paper should focus on the solution to the problem, thus adding more concrete information on the types of radiotherapy currently used for the palliation of bone metastasis, describing shortly the RT delivery, while also mentioning the highly efficient targeted radionuclide therapy for this purpose.

2.     In the section on hyperthermia, the authors should offer some general technical details regarding HT delivery, the optimal heating range, thermal measurements for uniform temperature control, etc.

3.      “In combination with RT, HT may suppress bone pain in patients suffering from cancer with painful bone metastases, with minimal side effects [39].” – this is in contradiction with the next paragraphs where the authors list the adverse effects caused by the two therapies. Please rephrase or remove.

4.     The details about the phase III clinical trial are scarce. There are no details on the RT (technique, beam quality, energy) nor on the HT delivery. You mention as a limitation the small number of patients without indicating the number.

5.     Explain the treatment choice whereby ‘…and then mEHT was applied alone for two weeks’.

Materials and methods:

1.     There is a lot of background information in the first few paragraphs, overlapping with or complementing the info supplied in the Introduction. All generic information of the effect of HT on cells should be moved to Introduction.

2.     The case reports should not be in the Methods as they are not part of your methodology. Move that section to Discussion.

3.     Divide the Methods section into: (1) Patient selection; (2) RT protocol; (3) HT protocol; (4) Combined treatment protocol; (5) Data analysis, and re-arrange the paragraphs accordingly.

4.     Give more details on the RT protocol – how was the treatment delivered (SBRT via IMRT/VMAT or other technique? what was the photon energy?)

5.     The section on adverse effects (lines 249 – 253) is not part of methods. This is already a result; therefore, it should be moved accordingly.

Results:

1.     The first paragraph is not your results! This should be moved to Discussion, when comparing your results with the literature.

Discussion:

1.     Figures 7 and 8 are derived from your results, thus should be moved to that section.

2.     The Discussion lacks a comparative analysis between the author’s results and literature reports.

Conclusions:

1.     The conclusions should be a 1-2 sentence summary of the overall results and must not include new data, let alone new graphs. Please move figure 8 to Results!

2.     The limitations of the study must be part of Discussion and not Conclusions.

Specific comments:

1.     Abstract: for better clarity, replace “This retrospective study included three groups of 61 patients” with “This retrospective study included 61 patients divided into three groups”.

2.     Line 72 – replace ‘dramatically reduce’ with ‘dramatically reduced’

3.     Line 81 – replace ‘fractioning the RT’ with ‘fractionating the RT dose’. Replace ‘fractioning’ with ‘fractionating’ throughout the paper.

4.     Remove section from lines 90-92 (‘Bone metastases can lead…..overall comfort) as it is too repetitive.

5.     Line 124 – remove the sentence ‘HT and modern radiology are synergistic [34,51].’ It is highly repetitive and also incorrect. ‘Modern radiology’ refers to imaging techniques and not therapeutic ones.

6.     Line 145 – ‘hypoxia potentially resulting from HT’ – how does this occur? The sentence is not clear and even contradictory.

7.     Please correct reference # 27 in the reference list. Add all publication details such as journal name, issue, pages, etc.

Comments on the Quality of English Language

some minor editing is required

Author Response

Introduction:

  1. The authors put too much emphasis on the pain induced by bone cancer, which becomes highly repetitive throughout the paragraphs. The paper should focus on the solution to the problem, thus adding more concrete information on the types of radiotherapy currently used for the palliation of bone metastasis, describing shortly the RT delivery, while also mentioning the highly efficient targeted radionuclide therapy for this purpose.

As suggested by the reviewer, duplicate content in the pain caused by bone meta section in the introduction was removed and shortened.

  1. In the section on hyperthermia, the authors should offer some general technical details regarding HT delivery, the optimal heating range, thermal measurements for uniform temperature control, etc.

In the explanation of radiation therapy and hyperthermia, setting the patient's treatment range was explained in more detail.

Hyperthermia

The mEHT treatment was performed with an EHY2000+ device (Oncotherm, Troisdorf, Germany) 3 times/week for 4 weeks. A 2D projection of the 3D simulation was used to facilitate accurate positioning of the 30-cm circular mEHT applicator. Prior to modulated-electro hyperthermia, all patients underwent a two-dimensional simulation by Acuity (Varian Medical System, Palo Alto. CA, USA) . The treatment field encompassed the mass with a 3 cm margin in the X and Y directions. The body temperature, blood pressure and pulse rate of each patient were measured prior to, during and following treatment. Body temperature was measured using an infrared ear thermometer (Infrared Thermometer IRT 4020; Braun GmbH, Kronberg, Germany) and the temperature of the treatment area skin surface below the circular upper electrode probe was measured using a non-contact infrared thermometer transmitter (Thermo Checker DT-060; Easytem Co., Ltd., Siheung, Korea). 

We add the results temperature measurement results.

The modulated electro hyperthermia alone group, body temperature before the treatment ranged from 36.4 – 36.9 (mean 36.5) and that after treatment ranged fron 36.9 – 37.8 (mean 37.4he treatment area skin surface underneath the upper electrode before heating ranged from 35.2-36.1 (mean 35.3) and it increased to 39-40.5. The radiotherapy combined modulated electro hyperthermia group, body temperature before the treatment ranged from 36.2 – 36.7 (mean 36.4) and that after treatment ranged from 36.8 – 37.8 (mean 37.4he treatment area skin surface underneath the upper electrode before heating ranged from 35.0-36.1 (mean 35.2) and it increased to 39-40.4, indicating an average rise of 0.8

  1. “In combination with RT, HT may suppress bone pain in patients suffering from cancer with painful bone metastases, with minimal side effects [39].” – this is in contradiction with the next paragraphs where the authors list the adverse effects caused by the two therapies. Please rephrase or remove.

We agree that the sentence you pointed out contradicts the content of the next paragraph. We decided to delete that sentence.

  1. The details about the phase III clinical trial are scarce. There are no details on the RT (technique, beam quality, energy) nor on the HT delivery. You mention as a limitation the small number of patients without indicating the number.

In the explanation of radiation therapy and hyperthermia, setting the patient's treatment range was explained in more detail.

Radiation therapy

After immobilization and CT simulation, the target volume was defined after registration of the diagnostic magnetic resonance imaging, CT, or positron emission tomography/CT scan to the simulation CT scan (5-mm slice thickness). The affected bony lesions, including soft tissue parts, were delineated as the gross tumor volume. For coverage of areas with potential microscopic disease, a safety radical margin at least 20 mm around the gross tumor volume was defined as the clinical target volume. The planning target volume was generated with a 3-dimensional margin of 5 mm around the clinical target volume. Radiation therapy was performed using 10MV photon energy. All patients underwent three-dimensional conformal radiotherapy. Three-dimensional conformal RT was generated from a Clinic IX (Varian Medical System, Palo Alto. CA, USA) machine by the EClipse treatment planning system (Varian Medical System, Palo Alto. CA, USA).

  1. Explain the treatment choice whereby ‘…and then mEHT was applied alone for two weeks’.

Because lesion control continued for 2-3 weeks even after radiation treatment was completed, hyperthermia treatment was continued for 2 more weeks to increase the treatment effect during this period.

Materials and methods:

  1. There is a lot of background information in the first few paragraphs, overlapping with or complementing the info supplied in the Introduction. All generic information of the effect of HT on cells should be moved to Introduction.

We agree that the content mentioned by the reviewer is not included in the material and method section. The above content has been relocated to the introduction.

  1. The case reports should not be in the Methods as they are not part of your methodology. Move that section to Discussion.

Case reports have also been relocated to discussion section.

  1. Divide the Methods section into: (1) Patient selection; (2) RT protocol; (3) HT protocol; (4) Combined treatment protocol; (5) Data analysis, and re-arrange the paragraphs accordingly.

Thank you for your kind review. We've rearranged this section as you recommended.

  1. Give more details on the RT protocol – how was the treatment delivered (SBRT via IMRT/VMAT or other technique? what was the photon energy?)

In the explanation of radiation therapy and hyperthermia, setting the patient's treatment range was explained in more detail.

Radiation therapy

After immobilization and CT simulation, the target volume was defined after registration of the diagnostic magnetic resonance imaging, CT, or positron emission tomography/CT scan to the simulation CT scan (5-mm slice thickness). The affected bony lesions, including soft tissue parts, were delineated as the gross tumor volume. For coverage of areas with potential microscopic disease, a safety radical margin at least 20 mm around the gross tumor volume was defined as the clinical target volume. The planning target volume was generated with a 3-dimensional margin of 5 mm around the clinical target volume. Radiation therapy was performed using 10MV photon energy. All patients underwent three-dimensional conformal radiotherapy. Three-dimensional conformal RT was generated from a Clinic IX ((Varian Medical System, Palo Alto. CA, USA) machine by the EClipse treatment planning system (Varian Medical System, Palo Alto. CA, USA).

  1. The section on adverse effects (lines 249 – 253) is not part of methods. This is already a result; therefore, it should be moved accordingly.

The section the reviewer pointed out have also been relocated to result section.

Results:

  1. The first paragraph is not your results! This should be moved to Discussion, when comparing your results with the literature.

The paragraph the reviewer pointed out have also been relocated to discussion section.

Discussion:

  1. Figures 7 and 8 are derived from your results, thus should be moved to that section.

Thank you for your kind review. The Figures the reviewer pointed out have also been relocated to result section.

  1. The Discussion lacks a comparative analysis between the author’s results and literature reports.

We readjusted the case reports and we compared our data to first phase III clinical trials.

Conclusions:

  1. The conclusions should be a 1-2 sentence summary of the overall results and must not include new data, let alone new graphs. Please move figure 8 to Results!

As suggested by the reviewer, the conclusions have been shortened and concise.

  1. The limitations of the study must be part of Discussion and not Conclusions.

Thank you for your kind review. The part the reviewer pointed out have also been relocated to discussion section

Specific comments:

  1. Abstract: for better clarity, replace “This retrospective study included three groups of 61 patients” with “This retrospective study included 61 patients divided into three groups”.

We changed the sentence as the reviewer recommended.

  1. Line 72 – replace ‘dramatically reduce’ with ‘dramatically reduced’

We changed the sentence as the reviewer recommended.

  1. Line 81 – replace ‘fractioning the RT’ with ‘fractionating the RT dose’. Replace ‘fractioning’ with ‘fractionating’ throughout the paper.

We changed the sentence as the reviewer recommended.

  1. Remove section from lines 90-92 (‘Bone metastases can lead…..overall comfort) as it is too repetitive.

We removed the section.

  1. Line 124 – remove the sentence ‘HT and modern radiology are synergistic [34,51].’ It is highly repetitive and also incorrect. ‘Modern radiology’ refers to imaging techniques and not therapeutic ones.

we removed the sentence.

  1. Line 145 – ‘hypoxia potentially resulting from HT’ – how does this occur? The sentence is not clear and even contradictory.

As the reviewer recommended, we decided to remove the sentence.

  1. Please correct reference # 27 in the reference list. Add all publication details such as journal name, issue, pages, etc.

We added the publication details of the reference

  1. Ariyafar T, Mahdavi SR, Geraily G, Fadavi P, Farhood B, Najafi M, et al. Evaluating the effectiveness of combined radiotherapy and hyperthermia for the treatment response of patients with painful bony metastases: A phase 2 clinical trial. J Therm Biol. 2019, 84, 129-135.

Reviewer 3 Report

Comments and Suggestions for Authors

Hyperthermia has been used for long time and has been shown to be useful. However, it is rarely practiced today. This article is useful in reevaluating the use of hyperthermia. I also think it is a useful finding that hyperthermia, when combined with radiotherapy, has provided relief of pain due to bone metastases over a long period of time.

Page 6, line 231.

Because of the generation of heat shock proteins, I think that the usual frequency of hyperthermia therapy is once a week.

Why was it administered three times a week?

Also, which job position operated the hyperthermia equipment? (Physicians, nurses, radiologists, or other professionals?)

Author Response

Page 6, line 231. Because of the generation of heat shock proteins, I think that the usual frequency of hyperthermia therapy is once a week. Why was it administered three times a week?

Modulated electro-hyperthermia (mEHT) is an emerging non-invasive cancer therapy utilizing electromagnetic fields to selectively target cancer cells via temperature-dependent and independent mechanisms. However, mEHT triggers HSR in treated cells. However, the treatment target of modulated electro -hyperthermia is mainly cell membranes, and there are several reports that treatments targeting cell membranes show a slightly slower response. Since the main purpose of treatment for bone metastases is to control pain, it is performed 3 times a week to see the treatment effect a little faster and, when combined with radiation therapy, for a synergy effect with the radiation therapy effect. Hyperthermia was administered up to 3 times a week. However, there is a lack of actual research results on the extent to which HSP occurs when hyperthermia is administered once. Accordingly, there is a lack of research results regarding the appropriate heat treatment time. These researchers conducted short treatment intervals because the purpose of treatment was to relieve symptoms and control pain. It is believed that more prospective studies must be conducted to know the exact treatment interval, and the researchers also plan to conduct further experiments to provide guidelines for treatment times.

Also, which job position operated the hyperthermia equipment? (Physicians, nurses, radiologists, or other professionals?)

A radiation oncologist guided the overall operation of the hyperthermia treatment, and the operation of the equipment was performed by a radiologist under the supervision of a radiation oncologist.

Round 2

Reviewer 1 Report

Comments and Suggestions for Authors

The authors have satisfactorily addressed my previous comments, I have no further suggestions

Reviewer 2 Report

Comments and Suggestions for Authors

The authors have adequately addressed all comments raised by this reviewer.

Comments on the Quality of English Language

Minor corrections are required during galley proof.